# Anatomical and Phytochemical Characteristics of Different Parts of *Hypericum scabrum* L. Extracts, Essential Oils, and Their Antimicrobial Potential

**DOI:** 10.3390/molecules27041228

**Published:** 2022-02-11

**Authors:** Kubra Nalkiran Ergin, Songul Karakaya, Gamze Göger, Oksana Sytar, Betul Demirci, Hayri Duman

**Affiliations:** 1Department Pharmacognosy, Faculty of Pharmacy, Ataturk University, Erzurum 25240, Turkey; eczkubranalkiran@gmail.com; 2Department Pharmaceutical Botany, Faculty of Pharmacy, Ataturk University, Erzurum 25240, Turkey; 3Department Pharmacognosy, Faculty of Pharmacy, Trakya University, Edirne 22030, Turkey; gamzegoger@trakya.edu.tr; 4Institute of Plant and Environmental Sciences, Slovak Agricultural University in Nitra, 94976 Nitra, Slovakia; 5Department Pharmacognosy, Faculty of Pharmacy, Anadolu University, Eskisehir 26210, Turkey; betuldemirci@gmail.com; 6Department of Biology, Faculty of Science, Gazi University, Ankara 06560, Turkey; hduman@gazi.edu.tr

**Keywords:** antioxidant, antimicrobial, essential oil, *Hypericum scabrum*, a-pinene, secretory canals

## Abstract

*Hypericum* (Hypericaceae) is a genus that comprises a high number of species around the world. In this study, the roots, aerial parts, flowers, fruits, and aerial parts with flowers from *Hypericum scabrum* were macerated separately by methanol and water and then fractionated by different solvents of, such as ethyl acetate, *n*-hexane, butanol, dichloromethane, aqueous residue sub-extracts, and ethnobotanical use. All the extracts, sub-extracts and essential oils of *H. scabrum* were investigated for the first time in detail for their antimicrobial, total phenolics, and antioxidant activities. Anatomical structures of the root, stem, leaf, upper and lower leaf surface, stamen, sepal, and petal of *H. scabrum* were examined. The biochemical layout of essential oils was determined by GC and GC/MS. The antioxidant activity was determined by free radical scavenging activity (by DPPH). Antimicrobial activity was applied against *Candida albicans* ATCC 10231, *Escherichia coli* ATCC 8739, *Staphylococcus aureus* ATCC 6538, *Bacillus subtilis* ATCC 19659, and *C. tropicalis* ATCC 750 using microdilution methods. The essentials of the aerial parts, flower, and fruit are characterized by the presence of monoterpene hydrocarbons, whereas roots oil include alkanes. The GC-FID and GC-MS analysis showed that major components of roots, aerial parts, flowers, and fruits oils were undecane (66.1%); α-pinene (17.5%), γ-terpinene (17.4%), and α-thujene (16.9%); α-pinene (55.6%), α-thujene (10.9%), and γ-terpinene (7.7%); α-pinene (85.2%), respectively. The aerial part sub-extracts indicated a greater level of total phenolics and antioxidant potential. The *n*-hexane sub-extracts (from aerial part, flower, and aerial part with flower) showed the best activity against *B. subtilis*, with 39.06 µg/mL MIC value. The presented research work indicates that *H. scabrum* can be a novel promising resource of natural antioxidant and antimicrobial compounds.

## 1. Introduction

The medicinal use of *Hypericum* species dates back to 2400 years ago [1]. *Hypericum scabrum* L. belongs to the genus *Hypericum* (family Hypericaceae), which is known by the names of ‘mayasıl otu, kepir otu, kızılcık otu’ among the people in Anatolia, Turkey [2]. *H. scarbum* is also widespread in Afghanistan, Iran, Iraq, Syria, Turkey, Azerbaijan, Kazakhstan, Turkmenistan, Tajikistan, Uzbekistan, and Pakistan [3]. Turkey is a large center for the *Hypericum* genus [4]. In the Turkey region, *H. scabrum* can grow in many cities, such as Hakkari, Siirt, Kastamonu, Elazig, Erzurum, Sivas, Van, Bayburt, Antalya, and Ankara [2,5]. It is used by the public against constipation and hemorrhoids, including the infusion processed from its flowering branches [5,6].

*H. scabrum* is a perennial and hairless plant, and its stem is between 10–60 cm. Its leaves are opposite and entirely oblong-linear; its flowers are yellow, and the inflorescence is corimbus. Its calyxes are in the form of a permanent tube, with 5 pieces up to 3 mm, and its corollas have 5 petals. The stamens are plenty, and triadelphous and its fruits are septicide capsules. The seeds, on the other hand, are 2 mm long and covered within plenty light-colored papillae on the surface and are dark in color. This plant is an herbaceous plant pertaining to the *Hypericaceae* family that can grow at an altitude of 750–3200 m, growing in dry, rocky slopes, open woods, or steppe habitats [5,6,7,8,9]. In addition, its ethnobotanical use has been reported in many diseases, such as cystitis, heart diseases, and intestinal disorders [10]. It has been reported the high content of α-pinene in *H. scabrum* species [11].

It has been used in the treatment of disorders, such as heart and neurological diseases, rheumatism, jaundice, ulcers, gastritis, hemorrhoids, constipation, and bladder and bowel issues [10,12,13,14]. Recent studies on *H. scabrum* plant showed that it has cytotoxic [15], anti-inflammatory, antioxidant [16], antifungal [17], and antidepressant [18] activities. Today, a special interest in representatives of the *Hypericum* sp. specific compounds with antifungal, antibacterial, and antiproliferative effects need to be studied in detail, as well [14,19].

Antibacterial and antimicrobial drugs are some of the most frequently used drug groups in human treatments. The widespread use of antibiotics has improved the resistance mechanisms of bacteria and made bacteria-resistant pathogens [20]. With the development of the resistance mechanism, the useful life of antimicrobial drugs is shortened, but the need for drugs is increasing by the day [21]. The random usage of antibiotics has induced drug resistance in many bacterial strains, and the improvement of novel antimicrobial compounds for resistant organisms is of critical importance. However, the most significant resources of antibiotics are molds, actinomycetes, and bacteria, and higher plants also represent many classes of secondary metabolites with the character [22]. That is why scientists are researching plants as alternative antimicrobial agents. There are many aromatic compounds in *H. scabrum* plants. Just in 2020, the scientists found three new compounds with untypical skeletons that were segregated from the air-dried aerial parts of *H. scabrum* plant. The compounds revealed a moderate activity against *T. brucei* and *P. falciparum* [23]. These structures mostly consist of phenolic structure or oxygen-bound derivatives of these structures. Essential oils obtained from plants have been widely used as antiviral, acetylcholinesterase inhibitors, antibacterial agents, and fungicides, especially today [24,25,26]. It has been determined that essential oils, which have been examined by many researchers, have antimicrobial properties [27]. It was observed antimicrobial activity and synergy effects of *H. scabrum* essential oil with antifungal drugs [28]. In addition, the antioxidant and antimicrobial effects of many isolated biochemical compounds from plant have been determined [28,29].

*Hypericum* species have different medicinal effects due to the secondary metabolites they contain from at least 11 different classes. Many of its pharmacological effects are due to the presence of essential oil of naphtodiantron compounds and flavonoids [1]. It has been reported a high level of α-pinene in *H. scabrum* species [11]. One of the most important structures responsible for the activity of natural antioxidants is phenolic compounds. Phenolic compounds were identified in plants, such as coumarins, cinnamic acid derivatives, and flavonoids [29]. Flavones, xanthones, flavonoids, and essential oils were established in *Hypericum* species [30]. The findings from the other investigation showed that the main compounds of *H. scabrum* oil were found as α-pinene, spathulenol, β-pinene, α-cadinol, limonene, and epi-α-muurolol has antioxidant activity [31].

In the presented study, firstly investigated in detail were biological activities and biochemical (spectrometrical and GC and GC/MS methods) and anatomical structure of the different parts of *H. scabrum* plant. Antimicrobial and antioxidant activity of extracts (methanol (MeOH), water), sub-extracts (hexane, dichloromethane, ethyl acetate, butanol, water), oil mixture, and essential oils obtained from various parts of *H. scabrum* were evaluated.

## 2. Results

The amounts and yields of the extracts and sub-extracts of *H. scabrum* different parts are given in Table 1. The highest yield was obtained from MeOH extract of flowers, whereas the least yield was obtained from hexane sub-extract of roots.

The yields of the essential oils and colors are given in Table 2. The color of essential oils from the aerial part, flower, and fruit were light yellow, whereas the root color was dark yellow. The highest yield was detected in fruit essential oil, whereas the least yield was detected in root essential oil.

The essential oils composition of *H. scabrum*, for different parts, is given in Table 3. A total of twenty-seven compounds, totaling 86.9%, of the oil were established in the essential oil of *H. scabrum* roots. Undecane was the major compound of the root oil amounting to 66.1%. The investigation of the aerial part of *H. scabrum* resulted in the determining of thirty-four essential compounds totaling 98.4% of the oil. α-Pinene, at 17.5%, was the most abundant compound in the essential oil, followed by γ-terpinene (17.4%) and α-thujene (16.9%). Thirty-nine compounds were characterized in the oil of the flowers of *H. scabrum*, finding 99.5% of the oil. The main components were found to be as α-pinene (55.6%), α-thujene (10.9%), and γ-terpinene (7.7%). The analysis on the fruits of *H. scabrum* resulted in the detection of thirty-six essential compounds, totaling 99.6%. α -Pinene, at 85.2%, was the most abundant compound in the fruit oil. The detected compounds were categorized mainly as monoterpene hydrocarbons in aerial part (78.0%), flower (94.2%), and fruit (97.9%), whereas alkanes were the most abundant group in root oil.

The findings of content of total phenolics from specimens are indicated in Table 4.

The higher level of total phenolic was found in aqueous residue and dichloromethane sub-extracts of aerial part (94.850 ± 0.074 and 81.443 ± 0.225 mg/g, respectively), whereas the least content of phenolics was detected in essential oils from root and fruit (2.47 ± 0.112 and 4.997 ± 0.321 mg/g, respectively).

The results of DPPH radical scavenging activity of the extracts, sub-extracts, essential oils, and ethnobotanical use from *H. scabrum* are presented in Table 5. The aqueous residue and dichloromethane sub-extracts of aerial parts (92.45 + 2.84 and 90.31 + 5.12 μg/mL, respectively) had the highest radical scavenging effect when compared the standards propyl gallate and rutin. However, essential oils of root and fruit (21.25 + 1.45 and 25.31 + 1.34 µg/mL, respectively) presented the lowest DPPH radical scavenging activities. At the same time, inhibition above 70–80% may be out of the linear range of dependence I(%) versus C(ug/mL), and it is cannot be correctly compared with standard solutions.

Minimum inhibitory concentrations value (μg/mL) of the extracts (MeOH and lyophilized extracts, aqueous, *n*-hexane, dichloromethane, EtOAc, BuOH, and aqueous residue) sub-extracts, essential oils, and ethnobotanical use form from *H. scabrum* and standard antimicrobials drugs are given in Table 6. 

Generally, the root extracts of MeOH, *n*-hexane, and dichloromethane from *H. scabrum* were found to be effective against *B. subtilis*, *C. albicans*, and *C. tropicalis* with MIC = 156.25 µg/mL. The same extracts were also showed an inhibitory effect against *S. aureus* with MIC = 312.5–625 µg/mL. The essential oil from roots was not found effective against tested microorganisms with MIC ≥ 2500 µg/mL.

The aerial part extract of *n*-hexane from *H. scabrum* showed the best activity against *S. aureus* and *B. subtilis* with MIC = 39.06 µg/mL. The aerial part extract of EtOAc was found to be effective against *C. tropicalis* with MIC = 312.5 µg/mL. The essential oil from aerial part was only found effective against *C. tropicalis* with MIC = 156.25 µg/mL.

The flower extract of *n*-hexane from *H. scabrum* showed the best activity against *B. subtilis* with MIC = 39.06 µg/mL. The flower extracts of MeOH, *n*-hexane, dichloromethane, EtOAc, and aqueous residue from *H. scabrum* were found effective against *S. aureus*, *B. subtilis*, *C. albicans*, and *C. tropicalis* with MIC = 156.25 µg/mL. The essential oil obtained from flower was found active against *C. tropicalis* with MIC = 312.5 µg/mL.

The fruit extracts of EtOAc, BuOH and aqueous residue from *H. scabrum* were found effective only for *C. albicans* with MIC = 156.25 µg/mL.

The aerial part with flower extracts of MeOH and *n*-hexane from *H. scabrum* showed lower MIC inhibition with 39.06 µg/mL against *S. aureus* and *B. subtilis*. Furthermore, EtOAc extract was only found effective for *C. tropicalis* with MIC = 156.25 µg/mL. The ethnobotanical uses of *H. scabrum* were only found effective for *C. tropicalis* with MIC = 156.25 µg/mL.

Root anatomy: *H. scabrum* root cross-section was examined with both chloralhydrate and sartur reagents. In the root cylindrical structure, there is the outermost exoderm layer. The periderm layer was observed under the exoderm layer. After the peridermal layer is the cortex parenchyma. Secretory canals were found in the cortex parenchyma. Secretory channels were also observed between the cortex parenchyma and the phloem. As it progresses toward the center, there are phloem and xylem layers, respectively, but there are trachea and tracheitis in the xylem layer (Figure 1).

Stem anatomy: A single layer of the epidermis was observed in the outermost part. Epidermis cells were observed in the rectangular structures. Under the epidermis layer, there is an average of several rows of parenchyma, which contains a large number of starches that are stained blue with Sartur reagent. A single layer of endodermis was observed after the cortex parenchyma. Starches stained blue with Sartur reagent were found in the endodermis layer. There are many secretory canals between the phloem and the cortex parenchyma. Starches were found in this region (Figure 2).

Leaf cross-section anatomy:

Leaf Lower Superficial: There are stomata on the lower surface of the leaf. When the neighboring cells of the stoma are examined, they show a rectangular shape and a chain-like appearance. The stoma type is anisostic (Figure 3).

Leaf Midrib: When the midrib of the leaf and the general cross-section of the leaf are examined, there is a single layer of epidermis at the bottom and the top. Stomata were observed in the upper and lower epidermis. Palisade parenchyma is present under the epidermis layer. Upper palisade parenchyma was observed in 3 rows and lower palisade parenchyma in 2 rows cylindrical structure (Figure 4).

Leaf Upper Superficial: Abundant stomata were observed on the upper surface of the leaf. The cells adjacent to the stoma are anisostic. Stomata were observed on both the upper and lower surfaces of the leaf. It has amphistomatic leaves (Figure 5).

Stamen Anatomy: When the stamen structure was examined, a large amount of druse was found in the anther structure. Pollens were observed (Figure 6).

Sepal Anatomy: When the sepal structure was examined, glandular growths, which are important in terms of diagnosis, were observed. Secretory growths vary in number between 12 and 16 at the apex and on both sides of the apex. The secretion is ovalized and enlarged toward the top of the growths, and its color varies between yellow and orange. Between the veins on the surface of the sepal, there are a large number of round secretory structures of bright yellow color. A large amount of starch stained blue with Sartur reagent was found around the sepal. Sclerenchyma structure was observed (Figure 7).

Petal Anatomy: When the petal structure was examined, secretion structures were observed. There are 10 or more secretory growths at the apex of the petal. The apex of these glandular projections is oval-shaped and yellow-orange in color. Xylems are prominently and widely observed in the petal structure. Pollen was abundant. Starch grains stained blue with Sartur reagent were found (Figure 8).

## 3. Discussion

In this study, the anatomical structures of the *H. scabrum* belonging to the *Hypericaceae* family, which is widely grown in Turkey and collected from Erzurum province, were examined. In addition, the chemical structure of essential oils from different parts of the plant, the phenolic contents, and the antioxidant and antimicrobial properties of different extracts were analyzed.

Essential oil analysis of different parts of *H. scabrum* was performed with GC and GC/MS. In our study, the most essential oil yield was obtained from the fruit (1.299%) part, and the least from the root (0.003%) part. The order of essential oil yield is fruit > flower > aerial part > root. In a study by Heshmati et al. the essential oil yield obtained from *H. scabrum* flower was found to be 0.35% [16]. In another study, the essential oil yield obtained from the aerial part of the plant was determined as 0.4% [11]. In another study conducted in 2021, the essential oil yield obtained by hydrodistillation from the aerial part of *H. scabrum* was found to be 0.6% [28]. Comparative analysis with the literature and our results shown that results are compatible, but the essential oil yield of the root part remained at a very low level. There are no studies in the literature in which the yield of essential oil obtained from the root part of the plant is calculated.

In the research work of Khorshidi et al. (2020), α-pinene, β-pinene, limonene, and E-caryophyllene were estimated as the main components of *H. scabrum* [32]. In the study by Heshmati et al. α-pinene (46.3%) was found as the major component in the essential oil obtained from the flowers of the plant [16]. In the study of Ghasemi Pirbalouti et al. the most found compound in the essential oil of the flower was α-pinene (49.96%) [33]. In our study, α-pinene was found to be the major component at a rate of 55.6% in the essential oil of the flower. Comparative analysis of the flower essential oil of the literature and our studies was done and the highest level of α-pinene was found in our study, with a rate of 55.6%. The reason for this was thought to be effective when the flowers of the plant were collected, the way they were dried, and the localization where the plant was collected. In a study by Morteza-Semnani et al. the essential oil of the aerial part of the plant was analyzed and α-pinene was found in the highest amount at the rate of 45.3% [34]. Similarly, in the study of Sharopov et al. α-pinene was found to be 44.8% [11]. In another study, in which essential oil analysis was performed from the aerial parts, α-pinene remained at the level of 18.88% [6]. In another study conducted in 2021, the major component α-pinene was found to be 37.8% in the aerial part essential oil [28]. In our study, essential oil analysis of the fruit and root part was made.

In the study carried out by Heshmati et al. in 2018, the maximum amount of phenolic substance in essential oil, water, methanol, ethanol, ethyl acetate, and acetone extracts prepared from flowers was investigated, and the methanol extract had the highest phenolics (95.65 ± 4.72 µg GAE/g) [16]. In our study, 76.27 ± 2.59 mg GAE/g was calculated in the methanol extract prepared from flowers. In our study, the amount of phenolic contents closest to this result was observed in the aqueous residual sub-extract (94.85 ± 0.07 mg GAE/g) obtained from the aerial part (leaf + stem). The ethanol extracts from aerial parts of the three *Hypericum* species (*H. perforatum*, *H. scabrum*, and *H. origanifolium*) from Turkey were studied and exhibited strong DPPH and total antioxidant activity [35]. In our study, EtOAc extract showed a greater level of phenolics and antioxidant activity, as well compared to the ethnobotanical used essential oil of *H. scabrum*.

If we look at the antioxidant studies in the literature, Heshmati et al. compared the DPPH radical scavenging capacity of acetone, ethyl acetate, ethanol, methanol, water extracts, and essential oil of the flowers of the plant. The highest activity was detected in the methanol extract, and the lowest activity was observed in the water extract [16]. In our study, the highest antioxidant activity was observed in methanol extract of flower, and it was found to be compatible with the study. In the study of Keser et al. when the DPPH radical scavenging activity of water and ethanol extracts obtained from flowers was examined, the scavenging activity of the water extract was 91.66%, and the DPPH scavenging activity of the standard substance BHA was 90.16%, and it showed better activity [36]. The ethanol extract was not examined in our study, and the aqueous residue sub-extract of the flower (73%) gave better results than our reference rutin (71%). The study of Shafaghat investigated the DPPH radical scavenging effect of hexane extracts of the leaves, flowers, stems, and seeds of the plant. The significant antioxidant activity was observed in the extracts of seeds, leaves, flowers, and stems. At the same time, antioxidant activity of extracts did not show better results than standard vitamin C [37]. Seeds extracts were not examined in current study. Our results showed better radical scavenging activity than our standards.

In a study by Abdollahi et al. in 2012, the effects of ethyl acetate and aqueous extract received from flowers and leaves of *H. scabrum* species against four Gram positive, three Gram negative, and three fungal strains were investigated, and inhibition zones were measured. Although the antimicrobial effect was observed on both bacterial and fungal strains, it was found ineffective against *E. faecalis* bacteria [17].

In a study conducted in 2012, the antibacterial and antifungal effects of hexane extracts of seeds and leaves of *H. scabrum* were researched against four Gram-positive, three Gram negative, and three fungal strains. Results of studied all hexane extracts, exhibited moderate activity, except *A. niger* fungus, *K. pneumoniae*, and *P. aeruginosa* bacteria. The highest effect was observed in *E. coli* and *B. subtilis* [38]. In a study by Keser et al. water and ethanol extracts of *H. scabrum* flowers were investigated against *E. coli*, *P. vulgaris*, *P. aeruginosa*, *L. monocytogenes*, *K. pneumonia*, *B. subtilis*, *B. megaterium*, *S. aureus*, and *C. albicans*. All extracts showed stronger activity than the reference agents streptomycin and nystatin [36]. In the previous studies, plant extracts were found to be effective against the Gram-negative bacteria *E. coli.* In our study, methanol and hexane extracts of the root parts of *E. coli* were found to be moderately effective. No effect was observed against *E. coli* by essential oils.

In our study of *S. aureus*, the hexane extract of the aerial part and aerial part with flower extracts showed the best results, while the extract of aerial part with flower and flower extract showed the best results against *B. subtilis.*

These results were found to be compatible with the literature. In literature studies, especially flower extracts of the plant showed strong antimicrobial activity against Gram-positive bacteria. In our study, the best antimicrobial results in the extracts were found against *S. aureus* and *B. subtilis* (MIC value 39.06 µg/mL). In antifungal activities, however, no extracts have been shown better antifungal activity than the fluconazole and terbinafine in our study. However, methanol, hexane, dichloromethane extract of the root structure of the plant to *C. albicans,* hexane extract of flower structure, ethyl acetate, butanol, aqueous residue extracts of fruit parts (MIC value 156.25 µg/mL) showed good antifungal activity. There are also studies in the literature investigating the antifungal activity of flower extracts against *C. albicans* species.The inhibition zone of the reference (Nystatin) was found to be 10 mm. Inhibition zone of water and ethanol extract of flower part against *C. albicans* fungus was measured as 18 mm and 25 mm [36]. The antifungal activity of root and fruit extracts of *H. scabrum* has not been presented in the literature.

Anatomical structure of root, stem, leaf, upper and lower leaf surface, stamen, sepal, and petal were examined. According to the source prepared by Metcalfe and Chalk, when the basic anatomical structure of the *Hypericaceae* family is examined, the most characteristic features are the schizogenic secretory spaces, which are observed as translucent or opaque spots in the leaves of all genera. The leaves are usually dorsiventral. There are secretory channels in the phloem, petiole, veins, and sometimes in the primary cortex, and the number of these secretory channels is important. Phloem and xylem, which form closed rings especially in herbaceous species, are generally narrow in structure, and the core region of these species is more developed. It is usually papillose in the genus *Hypericum*. Stomata are surrounded by 3 or more cells and are usually limited to the lower surface of the leaf. Some species have hypoderm. The endodermis layer is well defined in the young stem layer of the genus *Hypericum*, especially in terrestrial regions. The mesophyll usually contains a single palisade tissue, although, in some species, it contains two palisade tissues [39]. In the thesis study prepared by Ebru Yüce in 2009, the epidermis layer consisting of a single row of cells and an exodermal layer with 4–5 rows were seen in the cross-section of the *H. scabrum* root. The cortex layer is followed by endoderm, phloem, and xylem. Cambium layer was seen between phloem and xylem. In the transverse section of the stem, the outermost cuticle layer, epidermis going toward periderm, cortex parenchyma, sclerenchyma tissue, endodermal layer, phloem, cambium, xylem, and pith region was observed. Although the stomata cells are amaryllis type stomata, they are surrounded by three cells and are anisostic [40].

In another thesis study, prepared in 2019, as in other studies, root cross-section exoderm, 3–4 rows of periderm, cortex parenchyma, phloem, and xylem were observed. A single-layered epidermis, 4–7 rows of cortex parenchyma, thickened sclerenchyma tissue toward the center, and phloem, xylem, and pith after this tissue was seen in the transverse section of the trunk. Upper and lower epidermis were observed in the leaf structure. Lower and upper palisade parenchyma were seen under the epidermis layer. Secretory channels formed large gaps. Stomata are observed on the lower and upper surfaces and they are anamostic [41]. In a study by Polat et al. the cross-section of the stem and leaf of *H. scabrum* was examined. In the transverse section of the stem, a single layer of the epidermis, 1–2 layers of collenchyma under the epidermis, chlorenchyma, multilayered cortex parenchyma, endodermis, phloem, prominent pith rays, xylem, and pith region are observed. In the cross-section of the leaf, there is the epidermis layer and a thin cuticle is seen on this layer. The leaf was observed as equifacial. Upper and lower palisade parenchyma were seen. There is a collateral vein in the leaf midrib. Stomata was anisocytic, and leaves were amphistomatic [42]. If we compare our study with the literature study, results compatible with the literature were found.

When the root cross-section was examined, a 2-row exodermal layer was observed following the epidermis layer. The cortex parenchyma was observed as 3–4 layers. Secretory channels were found in the cortex and phloem layer. The phloem layer was followed by the xylem and pith region. In our study, when the cross-section of the stem structure is examined, the outermost layer of the epidermis consisting of a single row of rectangular cells is located. Under the epidermis layer, 4–5 rows of collenchyma in a round structure were observed. Under the collenchyma, 3–4 rows of cortex parenchyma, endoderm, phloem, trachea, and xylem layer containing tracheids and pith region were detected. Numerous secretory channels were found in the cortex parenchyma and phloem layer. The core region is well developed according to the phloem and xylem layer following the structure of the herbaceous species of the genus *Hypericum*. These secretory channels and pith region are in agreement with Metcalfe and Chalk’s [39] and other studies. In the literature, the wing structure was only observed in the number 3–5 in the study of Polat et al. [42] Calcium oxalate crystals were found in clusters in the core region.

In our study, when the cross-section of the leaf midrib, leaf lower surface, and upper surface of the leaf is examined, there are differences between the studies conducted, but our study is compatible with the literature. When the cross-section of the leaf was examined, schizogenic secretion spaces, which are the most characteristic feature of the species, were clearly observed. Secretory channels were observed. When the secretory channels are examined in shape, it is observed in a round structure and intense color. Based on the literature studies, it can be said that type C channels are dense. Type C secretory ducts are expressed as wide spaces bounded by one or more layers of densely colored and thin-walled secretory cells [41,42,43]. A single layer of epidermis was observed on the lower and upper surfaces of the leaves. There is 3-row palisade parenchyma under the upper epidermis and 2-row palisade parenchyma under the lower epidermis. A large sclerenchyma bundle was found in the midrib of the leaf. The leaf structure of the genus *Hypericum* is generally dorsiventral according to the source prepared by Metcalfe and Chalk [39]. In our study and in other *H. scabrum* studies, the leaf was observed as equifacial. There are stomata on both the upper and lower surfaces of the leaf. Stomata have 3 adjacent cells, and neighboring cells differ in size. Stomata were anisostic in our study, as in other studies. The evaluation was made regarding anatomical studies that examined stamen, sepal, and petal from other *Hypericum* species. In a study by Minarchenko et al. petal, sepal, and leaf of 6 *Hypericum* species were examined for diagnostic purposes. Sepals of *Hypericum* species mostly remain intact due to their small size. For this reason, the location and structure of the secretory ducts is important in determining the species-specific characteristics. During anatomical analysis of the sepals of *H. alpigenum*, *H. hirsutum*, *H. montanum*, and *H. elegans* found long, upwardly thickened secretory growths on the margins of the sepals. Large round, nearly black, dark-colored glands along or between the sepal midrib and numerous small yellowish round glands were seen in most species. For example, three types of secretion structures were observed in *H. elegans* species. These are oval thickening at the apex of the glandular outlets along the margin of the sepals, including the apex, yellow oval structures between the veins, and single dark round glands. These glandular growths provide information about the species [44]. In our study, round and broadly ovalized glandular growths were observed on the apex of the sepal and around the yellow-orange colored apex, the number of which varies between 12–16. Around the sepal vessels, especially in the areas close to the apex, a large amount of yellow round secretion structures was found. A large amount of starch was observed, with prominent vascular bundles and staining sartur to blue. Although it is compatible with studies on other *Hypericum* species, there are significant differences in secretion growth compared to *Hypericum* species studied as a microscope image. No black clothes were observed. Yellow-colored round glands were seen in larger structures and in abundance. Secretory growths were concentrated at and around the apex and were not uniformly distributed throughout the sepal. It is the first study in the literature to determine the sepal, petal, and stamen-derived characteristics of *H. scabrum*.

In the literature study on the petal and the position, shape, and color of the secretory structures are important for species identification, e.g., in *H. perforatum* species, long yellow secretion structures were found between the petal veins. In the *H. elegans* species, black glands on the petal edge can be said characteristically. In *H. maculatum*, long black glands were found on the entire surface of the petal [44]. Analysis of the petal structure shown growths with secretory protrusions were of 10 or more in the apex part of the petal. The apex of these glandular growths was oval-shaped and yellow-orange. Small round yellow glandular structures were found in large numbers on the stem part of the glandular growths and on the surface of the petal. In addition, yellow-brown oval or elongated secretion channels were found between the veins of the petal. Plenty of pollens and sclerenchyma structures were seen in the petal structure.

Abundant druzes were found in another structure after examination of the stamen structure. Pollen and filament structure were observed. There are no studies in the literature examining the stamen structure of *H. scabrum*. In a study examining the stamen structure of *H. perforatum*, black glands were observed in the stamen structure [45]. In stamen examination of the current study of *H. scabrum*, no black glands were found in either anther and filament structure. As we see, the presented study tried to contribute to the scientific knowledges and literature by determining the characteristic features of the *Hypericum scabrum* plant, both phytochemically and anatomically.

## 4. Materials and Methods

### 4.1. Plant Materials

*H. scabrum* was first collected to be diagnosed in 2016, and the diagnosis was made by Dr. Hayri Duman. The study samples were collected from Palandöken Mountain near Abdurrahman Gazi Tomb and Erzurum Urban Forest between June and August 2019 and 2020. The plant was collected from the slopes of the mountain at an altitude of between 1910–1990 m at flowering and fruiting times. The Herbarium specimens have been conserved at the Biodiversity Application and Research Center of Atatürk University with the AUEF 1278 herbarium number. Plant specimens for activity experiments were dried in the shade, away from moisture. For anatomy studies, samples were taken into 70% ethanol during the collection of the plant.

### 4.2. Extraction and Fractionation

Different parts of *Hypericum scabrum* were powdered separately, and they were macerated with methanol at room temperature for 8 h/3 days with a mechanical mixer. Extracts evaporated to dryness were dispersed in a mixture of methanol:water (1:9), then fractionated with solvents of different polarities (*n*-hexane, dichloromethane, ethyl acetate, *n*-butanol), respectively, and the sub-extracts obtained by evaporating with a rotatory evaporator were weighed [46]. In addition, for the lyophilization of aqueous extracts, different parts of *H. scabrum* were powdered separately and subjected to maceration with a mechanical mixer with distilled water at room temperature for 8 h for 3 days. After the filtrates are frozen in a deep freezer at −40 °C, distilled water was evaporated with a lyophilizer, and the extracts were weighed.

### 4.3. Preparation of Ethnobotanical Use of H. scabrum

The fresh aerial part with the flower was mixed with olive oil in a transparent jar and was left in the sun for 40 days. After that, the mixture was filtered and taken in the vials. The most generally utilized topical preparation is *Hypericum* oil done from fresh or dried flowers or aerial parts with the flowers of the plant. For the preparation of oil, the sample is put in olive oil in transparent glass and kept in a warm place for fermentation. After that, the plant material is powdered, the oil is filtered, and then the plant is exposed to sunlight for about 4–6 weeks; along this time, the oil takes on intense red color [47].

### 4.4. Essential Oil Extraction and Analysis

The essential oils compositions from different parts of *H. scabrum* were analyzed. The plant parts were placed separately in balloons connected to a Clevenger appliance, water was added to them, and essential oils were obtained after approximately 3–4 h.

After the volumes of essential oils were read in the burette section, they were taken with glass Pasteur pipettes, and the yield calculation was made. The GC oven temperature was kept at 60 °C for 10 min, and 4 °C/min. The speed was also set to 220 °C, and, after being constant at 220 °C for 10 min, it was programmed to 240 °C at 1 °C/min incremental rate. The split ratio was programmed to 40:1, and the injector temperature to 250 °C. Mass spectra was recorded at 70 eV, and the mass range will be set between m/z 35 and 450. The FID detector temperature was set to 300 °C, and a second simultaneous automatic injection was performed on the same column by applying the same operating conditions to ensure the same elution order as GC-MS. The relative percentage amounts of the columns separated from the FID chromatograms were analyzed. Recognition of essential oil components was done by comparing the relative retention times of the original samples or by comparing the relative retention indices (RRI) with those of the *n*-alkanes series. Commercial libraries (Wiley GC/MS Library, MassFinder 3 Library) and the “Baser Essential Oil Components Library” created with original compounds and components of known oils, as well as computer matches against MS literature data, was used for identification systems.

### 4.5. Determination of Total Phenolics

The total phenolic content was established by using the Folin–Ciocalteu reagent following a little modified method [48]. The sub-extracts, extracts were evaporated and then dissolved with 500 μL of 70% methanol (HPLC-Gradient grade, Sigma–Aldrich, Darmstadt, Germany VWR chemicals) at 70 °C. The mixtures were centrifuged at 3500 g for 10 min, and the supernatants were collected in separate tubes. The pellets were re-extracted under identical conditions. Supernatants were combined and used for total phenolic assay. For the total phenolic assay, 20 μL of extract was dissolved into 2 mL of distilled water. Two hundred microliters of dissolved extract were mixed with 1 mL of Folin–Ciocalteu reagent (previously diluted tenfold with distilled water) and kept at 25 °C for 3–8 min; 0.8 mL of sodium bicarbonate (75 g L^−1^) solution was added to the mixture. After 60 min at 25 °C, absorbance was measured at 765 nm. The results were expressed as gallic acid equivalents. The measurement of total phenolics was done under absorbance at 765 nm with the use of a Jenway UV/Vis 6405 spectrophotometer (Jenway, Chelmsford, UK). The total phenolic contents were determined from the linear equation of a standard curve prepared with gallic acid. The results were calculated as mg of gallic acid equivalent per g dry weight of extract. 

### 4.6. Antioxidant Capacity

The DPPH radical scavenging activity of the samples was determined using the DPPH assay. The percentile of DPPH radical scavenging capacity was estimated by next formula: [(A_control_ − A_sample_)/A_control_] × 100, where A_control_ is the absorbance of the control reaction (all reagents except test compounds), and A_sample_ is the same absorbance [48,49].

### 4.7. Antimicrobial Activity

*Escherichia coli* ATCC 8739, *Staphylococcus aureus* ATCC 6538, *Bacillus subtilis* ATCC 19659, *Candida albicans* ATCC 10231, and *C. tropicalis* ATCC 750 were purchased from Microbiologics (San Diego, CA, USA). The antimicrobial activities of the extracts, sub-extracts, and essential oils were diluted two-fold initially with a final concentration range of 2500 to 39.06 µg/mL. Moxifloxacin, clarithromycin, cefuroxime, fluconazole, and terbinafine were prepared at 64–0.125 µg/mL within DMSO and water.

Antimicrobial activity tests were applied using microdilution methods for aerobic microorganisms (M-7-A7) and fungi (M-27-A3) published by the Clinical Laboratory Standards Institute (CLSI) [50,51].

Standard culture strains were stored at −85 °C until use. Cultures were seeded into petri dishes containing Mueller Hinton Agar MHA, Potato Dextrose Agar (PDA), or Sabroused Dextrose Agar (SDA) and incubated at 37 °C for 24 h. At the end of the incubation, it was taken from the single colonies growing on the medium and transferred to the tubes with Mueller Hinton Broth (MHB) (RPMI medium for Candida species) and incubated again at 37 °C for 24 h. Cultures grown in the broth, after 18–24 h of incubation, were adjusted for turbidity using a turbidimeter according to the McFarland No: 0.5 (approximately 10^8^ cfu/mL for bacteria, 10^6^ cfu/mL for yeast culture). Microplate petri dishes (Brand) with 96 “U”-type wells were used for the experiment. After these procedures, the lids of the microplate petri dishes were closed and incubated at 35–37 °C for 16–20 h. Microbial growth was noticed by adding 20 µL of resazurin of 0.01% with minor modifications of CLSI standards. It was then incubated for another 1–2 h at 37 °C for coloration. At the end of incubation, active bacterial cells reduce the resazurin (blue) to the resorufin (pink). The experiment was repeated 3 times, and then the mean of MIC was calculated.

### 4.8. Anatomical Studies

For anatomical studies, manual sections were taken from *H. scabrum* samples taken in 70% ethanol during plant collection, preparations were prepared with Sartur [52,53] and/or chloralhydrate reagents, and the preparations were photographed with a microscope connected to a computer. Images prepared with these reagents were recorded with a Zeiss 51425 camera attached to a light microscope (Zeiss 415500-1800-000, Carl Zeiss Microscopy, GmbH Konigsallee 9-21, 37081 Gottingen GERMANY). Preparations were prepared from the roots, stems, leaves, stamens, sepals, and petals, respectively, from the samples contained in alcohol. After the samples were placed in Styrofoam, manual sections were taken with a razor blade and placed on a slide with reagent dripped on it. Then, after being covered with a coverslip and heated, examinations were made.

### 4.9. Statistical Analysis

Whole data are imported as mean ± S.E., and variations between means were statistically analyzed by utilization of one-way ANOVA analysis, followed via Bonferroni’s complementary analysis, with *p* < 0.05 considered to indicate statistical expressiveness.

## 5. Conclusions

In the current study, the different plant parts of *H. scabrum* had a high presence of α-pinene, undecane, and α-thujene, and it can be used for the pharmaceutical industry. Undecane (66.1%) was found to be a major compound of the roots of *H. scabrum.* The highest level of a-pinene was found in the flower and fruit parts of *H. scabrum.* It was observed moderate antimicrobial effect against gram-positive bacteria of the extract, sub-extract, and essential oils of the *H. scabrum.* The extraction solution significantly influences the level of total phenolics and antioxidant potential in the studied extracts. EtOAc sub-extract showed a greater level of phenolics compared to the ethnobotanically-used essential oil of *H. scabrum*. Although the effect cannot be achieved in vitro as much as the antimicrobial agents in the market, it was suggested that it could be an alternative to the bacterial resŞistance mechanism, which is one of the biggest problems today, due to the major components in its content.

## Figures and Tables

**Figure 1 molecules-27-01228-f001:**
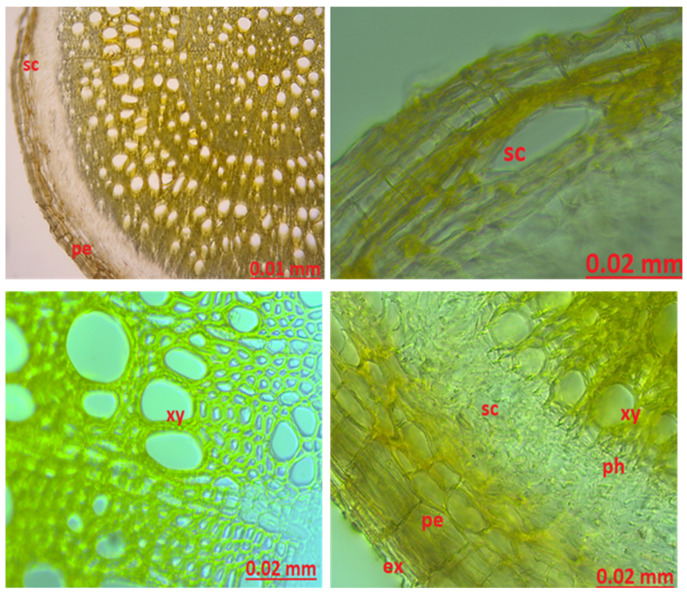
The root anatomy of *Hypericum scabrum*; ex: exoderm, pe: periderm, sc: secretory canals, ph: phloem, xy: xylem.

**Figure 2 molecules-27-01228-f002:**
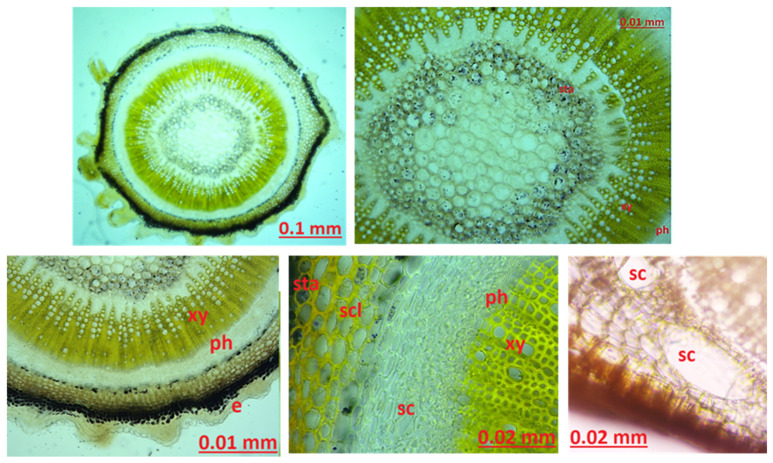
The stem anatomy of *Hypericum scabrum*; sc: secretory canals, ph: phloem, xy: xylem, scl: sclerenchyma, e: epidermis, sta: starch.

**Figure 3 molecules-27-01228-f003:**
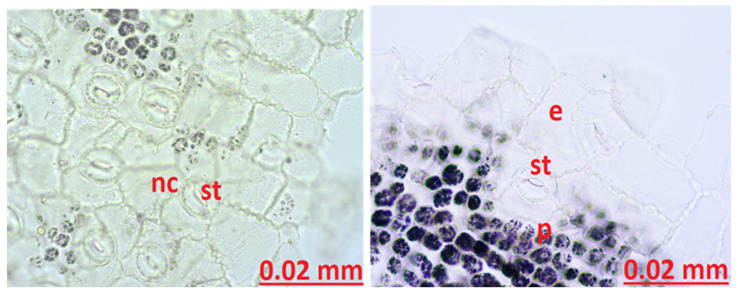
The leaf lower superficial anatomy of *Hypericum scabrum*; e: epidermis, p: parenchyma, nc: neighboring cells, st: stomata.

**Figure 4 molecules-27-01228-f004:**
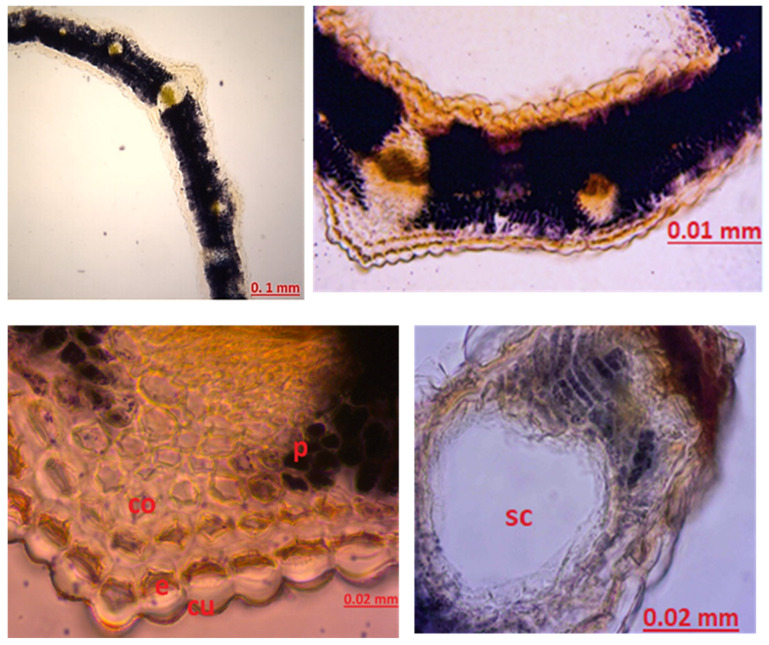
The leaf midrib transversal anatomy of *Hypericum*
*scabrum*; e: epidermis, p: parenchyma, sc: secretory canals, cu: cuticula, co: collenchyma.

**Figure 5 molecules-27-01228-f005:**
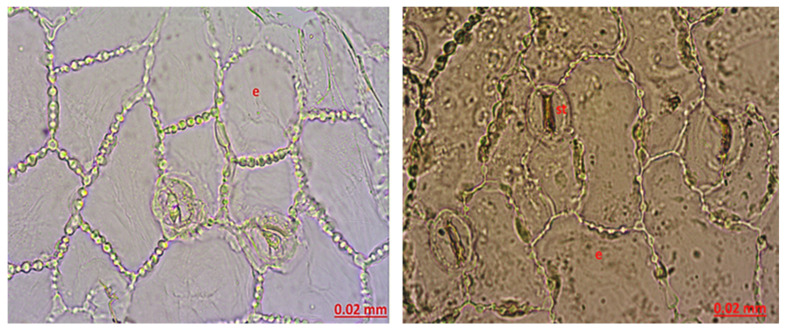
The leaf upper superficial anatomy of *Hypericum scabrum*; e: epidermis, st: stomata.

**Figure 6 molecules-27-01228-f006:**
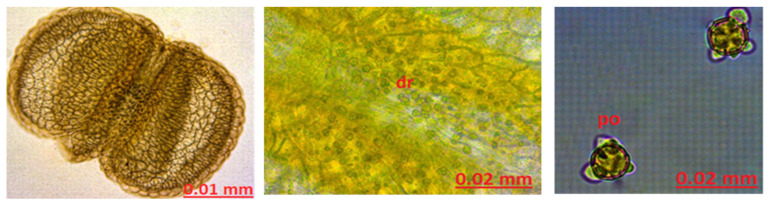
The stamen anatomy of *Hypericum scabrum*; dr: druse, po: pollen.

**Figure 7 molecules-27-01228-f007:**
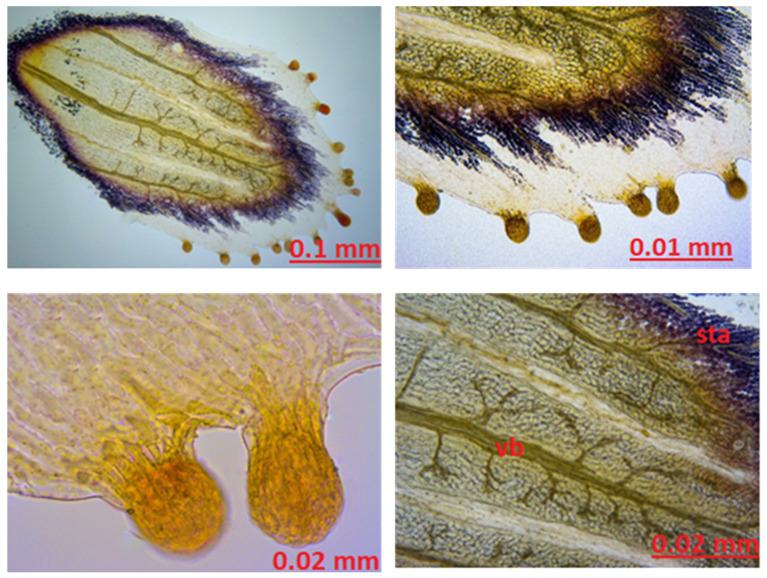
The stamen sepal of *Hypericum scabrum*; sta: starch, vb: vascular bundle.

**Figure 8 molecules-27-01228-f008:**
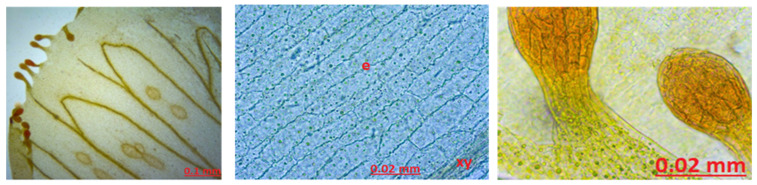
The stamen petal of *Hypericum scabrum*; e: epidermis, xy: xylem.

**Table 1 molecules-27-01228-t001:** The amounts (g) and yields (%) of the extracts and sub-extracts from different parts of *Hypericum scabrum*.

Used Parts	Root		Aerial Part		Flower		Fruit		Aerial Part with Flower	
	Amount (g)	Yield (%)	Amount (g)	Yield (%)	Amount (g)	Yield (%)	Amount (g)	Yield (%)	Amount (g)	Yield (%)
MeOH	7.10	10.15	21.94	24.94	29.28	40.11	15.65	23.36	19.88	21.05
Hexane (g)	0.41	0.59	1.83	2.09	2.92	4.00	3.02	4.51	1.03	2.99
Dichloromethane (g)	0.67	0.96	1.63	1.85	2.60	3.55	2.38	3.55	1.56	1.70
Ethyl acetate (g)	1.98	2.83	1.03	1.18	1.76	2.42	1.02	1.52	1.12	1.04
Butanol (g)	1.71	2.44	4.27	4.85	3.22	4.42	1.30	1.94	3.92	4.78
Aqueous residue (g)	1.40	2.01	8.14	9.25	12.89	17.66	4.37	6.51	6.56	7.01
Lyophilized Aqueous	2.56	5.02	9.10	11.68	14.90	20.96	6.80	8.89	8.91	9.09

**Table 2 molecules-27-01228-t002:** The yields of the essential oils and colours of *Hypericum scabrum*.

Used Parts	Plant Material (g)	Yield (% (*v*/*w*))	Color
Root	364	0.003	Dark yellow
Aerial part	332	0.301	Light yellow
Flower	340	0.588	Light yellow
Fruit	154	1.299	Light yellow

**Table 3 molecules-27-01228-t003:** The composition of the essential oils of *Hypericum scabrum*, for different parts.

RRI	Compound	Root %	Aerial Part %	Flower %	Fruit %	IM
1032	α-Pinene	1.4	17.5	55.6	85.2	RRI, MS
1035	α-Thujene	0.5	16.9	10.9	1.5	RRI, MS
1076	Camphene	-	-	0.1	0.1	RRI, MS
1100	Undecane	66.1	4.6	-	-	RRI, MS
1118	β-Pinene	-	2.0	4.1	4.4	RRI, MS
1132	Sabinene	-	4.2	3.2	0.4	RRI, MS
1174	Myrcene	-	2.0	2.5	2.1	RRI, MS
1176	α-Phellandrene	-	0.7	0.4	tr	RRI, MS
1188	α-Terpinene	tr	5.5	3.1	0.4	RRI, MS
1203	Limonene	0.2	2.0	1.1	1.0	RRI, MS
1218	β-Phellandrene	-	0.8	0.7	0.2	RRI, MS
1244	2-Pentyl furan	0.3	-		-	MS
1246	(*Z*)-β-Ocimene	-	tr	0.1	tr	MS
1255	γ-Terpinene	0.9	17.4	7.7	1.4	RRI, MS
1266	(*E*)-β-Ocimene	-	0.5	1.0	0.1	MS
1280	*p*-Cymene	0.6	5.7	2.4	0.8	RRI, MS
1290	Terpinolene	-	2.8	1.3	0.3	RRI, MS
1300	Tridecane	1.4	-	-	-	RRI, MS
1400	Tetradecane	0.2	-	-	-	RRI, MS
1492	Cyclosativene	1.6	-	-	-	MS
1497	α-Copaene	2.8	0.3	0.1	-	MS
1535	β-Bourbonene	-	-	tr	0.1	MS
1553	Linalool	-	-	tr	-	RRI, MS
1571	*trans-p*-Menth-2-en-1-ol	-	0.3	tr	tr	MS
1586	Pinocarvone	-	-	-	tr	RRI, MS
1589	β-Ylangene	-	-	0.2	-	MS
1597	β-Copaene	-	0.1	0.1	-	MS
1600	Hexadecane	0.4	-	-	-	RRI, MS
1611	Terpinen-4-ol	0.2	4.5	2.9	0.4	RRI, MS
1612	β-Caryophyllene	tr	tr	tr	tr	RRI, MS
1638	*cis-p*-Menth-2-en-1-ol	-	0.2	0.1	tr	MS
1648	Myrtenal	-	-	-	0.1	MS
1664	Nonanol	0.7	-	-	-	MS
1670	*trans*-Pinocarveol	-	-	-	0.1	MS
1683	*trans*-Verbenol	-	-	-	0.2	MS
1704	γ-Muurolene	2.6	1.5	0.4	0.1	MS
1706	α -Terpineol	-	0.3	0.2	0.2	RRI, MS
1725	Verbenone	-	-	-	0.1	MS
1726	Germacrene D	-	4.3	0.6	0.1	MS
1740	α-Muurolene	-	-	tr	-	MS
1755	Bicyclogermacrene	0.7	0.7	0.1	-	MS
1773	δ-Cadinene	2.3	1.5	0.3	0.1	MS
1776	γ-Cadinene	1.0	0.8	0.2	0.1	MS
1799	Cadina-1,4-diene (=*Cubenene*)	-	-	tr	-	MS
1804	Myrtenol	-	-	-	tr	MS
1807	α-Cadinene	tr	0.2	tr	-	MS
1827	(*E,E*)-2,4-Decadienal	0.3	-	-	-	MS
1845	*trans*-Carveol		-	-	0.1	RRI, MS
1849	Cuparene	1.9	-	-	-	MS
1849	Calamenene	0.3	0.3	tr	-	MS
1864	*p*-Cymen-8-ol		tr	tr	tr	MS
1868	(*E*)-Geranyl acetone	0.1	-	-	-	MS
1941	α-Calacorene	0.4	0.1	tr	-	MS
2144	Spathulenol	-	0.4	0.1	tr	MS
2187	T-Cadinol	-	tr	tr		MS
2250	α-Eudesmol	-	-	-	tr	MS
2255	α-Cadinol	-	0.3	tr		MS
2257	β-Eudesmol	-	-	-	tr	MS
2931	Hexadecanoic acid	tr	-	-	-	RRI, MS
	Monoterpene Hydrocarbones	3.6	78	94.2	97.9	
	Oxygenated Monoterpenes	0.2	5.3	3.2	1.2	
	Sesquiterpene Hydrocarbones	12	9.8	2.0	0.5	
	Oxygenated Sesquiterpenes	-	0.7	0.1	-	
	Alkanes	68.1	4.6	-	-	
	Fatty acid+esters	tr	-	-	-	
	Others	3.0	-		-	
	Total	86.9	98.4	99.5	99.6	

RRI: Relative retention indices; RRI: Relative retention indices calculated against *n*-alkanes; %: calculated from FID data; tr: Trace (<0.1%); IM: Identification method based on the relative retention indices (RRI) of authentic compounds on the HP Innowax column; MS, identified on the basis of computer matching of the mass spectra with those of the Wiley and MassFinder libraries and comparison with literature data.

**Table 4 molecules-27-01228-t004:** Total phenolic contents of the extracts, sub-extracts, essential oils, and ethnobotanical use from *Hypericum scabrum* (mg/g) ± SD ^a^.

Used Parts	Root	Aerial Part	Flower	Fruit	Aerial Part with Flower
MeOH	73.551 ± 4.98 ^a^	79.335 ± 0.221	76.272 ±2.591 ^a^	51.713 ± 0.449	65.487 ± 0.473 ^a^
Hexane	60.904 ± 0,37	76.664 ± 0.377	74.997 ± 0.147	52.105 ± 0.405	63.232 ± 0.368
Dichloromethane	54.703 ± 1.338 ^a^	81.443 ± 0.225 ^a^	68.300 ± 0.147 ^a^	51.884 ± 0.297	58.526 ± 0.321 ^a^
EtOAc	54.458 ± 0.112 ^a^	75.855 ± 0.225	60.193 ± 0.153	67.277 ± 0.258	77.767 ± 2.612 ^a^
BuOH	56.345 ± 0.112 ^a^	73.47 ± 0.112 ^a^	63.674 ± 1.977	56.026 ± 0.074	73.894 ± 0.074 ^a^
Aqueous residue	70.414 ± 0.634 ^a^	94.850 ± 0.074 ^a^	57.889 ± 0.779	10.708 ± 0.443	61.590 ± 0.612
Essential oil	2.47 ± 0.112	26.909 ± 0.555 ^a^	10.070 ± 0.147 ^a^	4.997 ± 0.321 ^a^	32.198 ± 0.309 ^a^
Ethnobotanical use					36,909 ± 0.512 ^a^

^a^ Standard deviation (*p* < 0.05).

**Table 5 molecules-27-01228-t005:** DPPH radical scavenging activity of the extracts, sub-extracts, essential oils, and ethnobotanical use from *Hypericum scabrum* (%).

Tested Samples	Percent DPPH Radical Scavenging Activity (%) ± SD ^a^				
	Root	Aerial Part	Flower	Fruit	Aerial Part with Flower
MeOH	85.21 + 4.21	88.34 + 4.32	87.95 + 5.12	68.75 + 2.56	82.67 + 4.53
Hexane	75.95 + 3.14	86.75 + 3.21	85.69 + 4.54	69.68 + 2.34	80.27 + 3.56
Dichloromethane	71.23 + 2.87	90.31 + 5.12	80.21 + 4.23	69.02 + 2.45	74.25 + 4.12
EtOAc	70.98 + 2.98	87.21 + 3.45	79.87 + 3.09	79.89 + 3.02	86.57 + 3.84
BuOH	72.87 + 3.05	84.56 + 2.14	83.26 + 3.75	75.25 + 2.87	85.87 + 3.84
Aqueous residue	82.69 + 1.89	92.45 + 2.84	73.12 + 2.16	45.68 + 2.35	79.45 + 1.69
Essential oil	21.25 + 1.45	50.89 + 2.12	43.12 + 2.45	25.31 + 1.34	62.28 + 1.72
Ethnobotanical use	57.89 + 3.56				
Propyl gallate	84.94 ± 0.73				
Rutin	71.09 ± 2.04				

^a^ Standard deviation (*p* < 0.05).

**Table 6 molecules-27-01228-t006:** The results of antimicrobial activities of the extracts, sub-extracts, essential oils, and ethnobotanical use from *Hypericum scabrum* and reference antibiotics (MIC values μg/mL).

Tested Samples		Microorganisms				
		*E. coli* ATCC 8739 (µg/mL)	*S. aureus* ATCC 6538 (µg/mL)	*B. subtilis* ATCC 19659 (µg/mL)	*C. albicans* ATCC 10231 (µg/mL)	*C. tropicalis* ATCC 750 (µg/mL)
Root	MeOH	625	312.5	156.25	156.25	156.25
	Hexane	625	312.5	156.25	156.25	156.25
	Dichloromethane	2500	625	156.25	156.25	156.25
	EtOAc	1250	2500	1250	1250	1250
	BuOH	1250	2500	2500	2500	2500
	Aqueous residue	1250	2500	2500	2500	2500
	Lyophilized aqueous	2500	>2500	>2500	2500	2500
	Essential oil	>2500	>2500	>2500	2500	2500
Aerial part	MeOH	2500	312.5	312.5	2500	1250
	Hexane	2500	39.06	39.06	625	625
	Dichloromethane	2500	312.5	156.25	>2500	1250
	EtOAc	2500	1250	1250	625	312.5
	BuOH	2500	>2500	>2500	2500	1250
	Aqueous residue	>2500	>2500	>2500	5000	2500
	Lyophilized aqueous	>2500	>2500	>2500	2500	625
	Essential oil	2500	2500	2500	1250	156.25
Flower	MeOH	2500	156.25	312.5	2500	312.5
	Hexane	2500	156.25	39.06	156.25	625
	Dichloromethane	2500	312.5	312.5	625	156.25
	EtOAc	625	312.5	312.5	312.5	156.25
	BuOH	>2500	312.5	>2500	1250	312.5
	Aqueous residue	>2500	156.25	>2500	>5000	1250
	Lyophilized aqueous	>2500	312.5	>2500	>2500	625
	Essential oil	>2500	>2500	>2500	2500	312.5
Fruit	MeOH	>2500	>2500	>2500	1250	625
	Hexane	>2500	2500	2500	625	2500
	Dichloromethane	2500	2500	2500	625	2500
	EtOAc	2500	2500	2500	156.25	312.5
	BuOH	1250	1250	1250	156.25	5000
	Aqueous residue	>2500	2500	2500	156.25	>5000
	Lyophilized aqueous	>2500	>2500	>2500	>2500	>2500
	Essential oil	>2500	>2500	2500	1250	2500
Aerial part with flower	MeOH	2500	39.06	78.12	312.5	625
	Hexane	2500	39.06	39.06	625	625
	Dichloromethane	2500	1250	1250	625	625
	EtOAc	1250	625	625	312.5	156.25
	BuOH	2500	1250	2500	2500	1250
	Aqueous residue	>2500	>2500	>2500	>5000	5000
	Lyophilized aqueous	2500	>2500	**625**	2500	>2500
Ethnobotanical use		>2500	2500	2500	2500	156.25
Clarithromycin		32	4	0.125	-	-
Fluconazole		-	-	-	0.25	0.5
Cefuroxime		32	32	>32	-	-
Terbinafine		-	-	-	4	8
Moxifloxacin		0.25>	0.25>	0.25>	-	-

## Data Availability

Not applicable.

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
