# Peer review of "Anatomical and Phytochemical Characteristics of Different Parts of Hypericum scabrum L. Extracts, Essential Oils, and Their Antimicrobial Potential"

_molecules, 2022, doi:10.3390/molecules27041228_

Round 1
Reviewer 1 Report
This study describes the antimicrobial and antioxidant activities of extracts and essential oils, essential oils profile, and anatomical properties of Hypericum scabrum L. (Hypericaceae). The topic of this study is rather interesting. Searching and analysis of new sources of potentially bioactive compounds are an actual trend of phytochemical studies. However, the essential oil profile and antimicrobial activity was also extensively studied in many previous studies on the plants of genus Hypericum. Discussion is well described and supported by results. Results and Discussion sections should be re-ordered to obtain a more systematic order. Conclusions summarize the most important findings, however, some drawbacks should be corrected. In my opinion, after a major revision of the manuscript, and some explanations, it can fulfil the basic requirements for publication.
Detailed comments:
Title: Antimicrobial and antioxidant activities of extracts, traditional use, essential oils, and anatomical properties of Hypericum scabrum L. (Hypericaceae). Traditional use was not the purpose of this study. I recommended removing “traditional use” from the title. Traditional use is not specified in the methodology. If it means traditional preparation of extract? If yes, it also is extracted, therefore “traditional use” should be removed - this meaning is misleading.
Line 17 – to which part of this phrase refers “and traditional use”? then fractionated by different solvents … and traditional use? This phrase is unclear. Rapharase or remove “and traditional use”.
Line 24 – “a slight modification” remove it. It is the detail for the methodology section.
Introduction
Line 25 – “Essential oils were obtained from the secretory canals of a plant. The canals of the aerial parts, flower, and fruit are defined by the presence of monoterpene hydrocarbons, whereas secretory canals of roots include alkanes.” Essential oils were obtained from plant material (different parts)… not strictly from the secretory canals.
Rather not „The canals” but essential oils of the aerial parts, flower, and fruit are characterized by the presence of monoterpene hydrocarbons, whereas roots include alkanes. Correct or rephrase.
Line 55 – “perennial” the same information as in line 47. Remove it.
Line 54 – “This plant is a….. „ move this phrase at the beginning of this paragraph (line 47)” as presents general information.
Line 93 – “The antioxidant activity of H. scabrum herb was higher compared to the Trolox and vitamin C standard and source of it was α-pinene [31].” The high antioxidant activity is mainly related to polyphenols. α-pinene is a weak antioxidant – it has no hydroxyl groups.
Results
In my opinion, this presented content in this section should be re-ordered as follows:
Fig. 1-8 and related results description should be at the end of this section (after Table 6) It allows to better transition between presented issues and results. Firstly, phytochemical properties, then anatomy.
Table 1 and 4. Unify the used terms – Methanol or MeOH. Use the same terms through all tables.
Table 1. Two decimal places are enough. Remove (g) after solvent. Add (g) after „Amount”. Remove vertical lines in the table.
Line 109 - The highest yield of essential oil was obtained for fruit, whereas the least for root?
Table 2 – “Powdered amounts” is the amount used for analysis? If yes, remove it. Yield (v/w) is expressed in % (v/w). or mL/g (or 100 g, or kg) of plant material? It is the yield per all powdered amount specified in the table as “Powdered amounts”, recalculate results and express as above - in % (v/w). or mL/g (or 100 g, or kg) of plant material.
Table 4-6 – specify “traditional use” meaning in methodology. This term is misleading. Traditional preparation? Describe how was obtained.
Discussion
Use the same order of presented issues as in results (following the described above re-order)
Firstly, discussion of phytochemical properties, then anatomy.
Line 473 – 477 – Why such a procedure was applied? Please, justify.
Why different parts of Hypericum scabrum were not extracted separately in methanol, water n-hexane, dichloromethane, ethyl acetate, and n-butanol? Extraction in methanol allows to separate only compounds soluble in methanol… (more polar), probably less polar compounds were not extracted, so further fractionation is limited only to more polar compounds (soluble in methanol). Extraction, for example with n-butanol, allow separating more polar compounds from the material than those separated by methanol. So, in my opinion extraction should be performed for each solvent separately.
„for 8 hours for 3 days” 8 hours or 3 days? Specify.
Line 515 – “dry weight” plant material or extract?. Specify.
Line 545 – any statistical analysis was performed? Experiments were performed in replicates (Table 1-3)?
Conclusions
Line 549 – „Hypericum scabrum has a very high antioxidant potential due high presence of a-pinene, undecane and a-thujene it can be used for the pharmaceutical industry.” As described above this statement is very questionable. Even results show that antioxidant activity of essential oil (where a-pinene is the main component of essential oils from aerial parts ) is significantly lower than in extracts. Extracts contains also polyphenols which are responsible for this activity. Consider the analysis of hyperoside and hypericin contents.
Line 555 – “EtOAc extract” strictly speaking EtOAc fraction of methanolic extract.
Author Response
Dear Reviewer,
Thank you very much for the all comments. It was helpful to improve MS.
The responces for all your comments you can find bellow.
- Title: Antimicrobial and antioxidant activities of extracts, traditional use, essential oils, and anatomical properties of Hypericum scabrum L. (Hypericaceae). Traditional use was not the purpose of this study. I recommended removing “traditional use” from the title. Traditional use is not specified in the methodology. If it means traditional preparation of extract? If yes, it also is extracted, therefore “traditional use” should be removed - this meaning is misleading
Dear reviewer, thank you very much for the comment. The article was revised, and some sentences were improved. The changes are in a red colour.
- Line 17 – to which part of this phrase refers “and traditional use”? then fractionated by different solvents … and traditional use? This phrase is unclear. Rapharase or remove “and traditional use”.
Dear reviewer, thank you very much for the comment.” Preparation of Ethnobotanical Use of Hypericum scabrum” was added at the material and methods part. The changes are in a red colour.
- Line 24 – “a slight modification” remove it. It is the detail for the methodology section. Introduction
Dear reviewer, thank you for the comment. “a slight modification” was removed.
- Line 25 – “Essential oils were obtained from the secretory canals of a plant. The canals of the aerial parts, flower, and fruit are defined by the presence of monoterpene hydrocarbons, whereas secretory canals of roots include alkanes.” Essential oils were obtained from plant material (different parts)… not strictly from the secretory canals.
Rather not „The canals” but essential oils of the aerial parts, flower, and fruit are characterized by the presence of monoterpene hydrocarbons, whereas roots include alkanes. Correct or rephrase.
Dear reviewer, thank you for the comment. It was rephrased.
- Line 55 – “perennial” the same information as in line 47. Remove it.
Dear reviewer, thank you for the comment. It was removed.
- Line 54 – “This plant is a….. „ move this phrase at the beginning of this paragraph (line 47)” as presents general information.
Dear reviewer, thank you for the comment. It was moved as the mentioned.
- Line 93 – “The antioxidant activity of H. scabrum herb was higher compared to the Trolox and vitamin C standard and source of it was α-pinene [31].” The high antioxidant activity is mainly related to polyphenols. α-pinene is a weak antioxidant – it has no hydroxyl groups.
Dear reviewer, thank you for the comment. This sentence was revised.
- Results
In my opinion, this presented content in this section should be re-ordered as follows:
Fig. 1-8 and related results description should be at the end of this section (after Table 6) It allows to better transition between presented issues and results. Firstly, phytochemical properties, then anatomy.
Table 1 and 4. Unify the used terms – Methanol or MeOH. Use the same terms through all tables.
Table 1. Two decimal places are enough. Remove (g) after solvent. Add (g) after „Amount”. Remove vertical lines in the table.
Line 109 - The highest yield of essential oil was obtained for fruit, whereas the least for root?
Table 2 – “Powdered amounts” is the amount used for analysis? If yes, remove it. Yield (v/w) is expressed in % (v/w). or mL/g (or 100 g, or kg) of plant material? It is the yield per all powdered amount specified in the table as “Powdered amounts”, recalculate results and express as above - in % (v/w). or mL/g (or 100 g, or kg) of plant material.
Table 4-6 – specify “traditional use” meaning in methodology. This term is misleading. Traditional preparation? Describe how was obtained.
Dear reviewer, thank you for the comment. The revisions were done.
- Discussion
Use the same order of presented issues as in results (following the described above re-order)
Firstly, discussion of phytochemical properties, then anatomy.
Dear reviewers, thank you for the comment. The revisions were done.
- Line 473 – 477 – Why such a procedure was applied? Please, justify.
Why different parts of Hypericum scabrum were not extracted separately in methanol, water n-hexane, dichloromethane, ethyl acetate, and n-butanol? Extraction in methanol allows to separate only compounds soluble in methanol… (more polar), probably less polar compounds were not extracted, so further fractionation is limited only to more polar compounds (soluble in methanol). Extraction, for example with n-butanol, allow separating more polar compounds from the material than those separated by methanol. So, in my opinion extraction should be performed for each solvent separately.
„for 8 hours for 3 days” 8 hours or 3 days? Specify.
Dear reviewer, thank you for the comment. The revisions were done.
- Line 515 – “dry weight” plant material or extract?. Specify.
Dear reviewer, thank you for the comment. The revisions were done.
- Line 545 – any statistical analysis was performed? Experiments were performed in replicates (Table 1-3)?
Dear reviewer, thank you for the comment. The statistical analysis was done for tables 4 and 5 and the Statistical analysis method was added.
-Conclusions
Line 549 – „Hypericum scabrum has a very high antioxidant potential due high presence of a-pinene, undecane and a-thujene it can be used for the pharmaceutical industry.” As described above this statement is very questionable. Even results show that antioxidant activity of essential oil (where a-pinene is the main component of essential oils from aerial parts ) is significantly lower than in extracts. Extracts contains also polyphenols which are responsible for this activity. Consider the analysis of hyperoside and hypericin contents.
Dear reviewer, thank you for the comment. It was revised. We agree about this sentence. We rewrote it as next “Hypericum scabrum has a high presence of a-pinene, undecane and a-thujene it can be used for the pharmaceutical industry”. We agree that antioxidant activity can be connected with presence of other antioxidants of phenolic nature and naphtodianthrones (hypericin etc.) as well. In our next studeis we will plan to estimate naphtodianthrones (hypericin etc.) content etc.
-Line 555 – “EtOAc extract” strictly speaking EtOAc fraction of methanolic extract.
Dear reviewer, thank you for the comment. It was revised.
Reviewer 2 Report
The title of the study requires a change and a clear expression of the focus of the text (eg the use of the phrase "traditional use" has no obvious meaning in the context of the title of the manuscript.
The text is not written uniformly, in many places there are inconsistencies - in various things - I give only an example - eg L4, L39, and L55 - the name of the family - italics.
In my opinion, the text is written in very poor English and needs to be revised by a native speaker.
Many sentences do not make sense and do not even have a sentence - see eg L69-71, L. 95-96 (the verb is completely missing ...). Again, I give only examples, but in many places, the text is incomprehensible or perhaps misleading. It must be revised, rewritten.
The same applies to object names - see eg Table 1 - the name is not well expressed, incorrectly in English.
The tables do not follow the instructions for the authors, the same applies to other parts of the manuscript. By the way, the individual categories in the tables are not completely named/expressed.
What is the significance of the reported "anatomy" for drawing the conclusions of the study? The objects listed in the text are not clearly described and therefore a clear understanding of the pictorial appendix is ​​not possible.
Table 4 - "traditional use" - what does it mean in the context of the data in the table?
Table 6 - It is not clearly specified that this is eg MeOH root extract of the material, etc. !! Everything needs to be reworked. Has the MIC of the prepared samples actually been found to be up to 80 times higher than that of commercial ATBs?
MM section - methodologies are not clearly and completely described (eg 4.1 - the origin of samples, location, collection time, etc. are not given at all; 4.4. - completely insufficiently described, it is not possible to repeat or find out the implementation) - the comment applies to all methodologies.
Author Response
Dear Reviewer,
Thank you for the useful comments and suggestions which were helpful to improve MS.
We also prepared responces for all comments which you can see bellow.
- Title: Antimicrobial and antioxidant activities of extracts, traditional use, essential oils, and anatomical properties of Hypericum scabrum L. (Hypericaceae). Traditional use was not the purpose of this study. I recommended removing “traditional use” from the title. Traditional use is not specified in the methodology. If it means traditional preparation of extract? If yes, it also is extracted, therefore “traditional use” should be removed - this meaning is misleading
Dear reviewer, thank you very much for the comment. The article was revised, and some sentences were improved. The changes are in a red colour.
- Line 17 – to which part of this phrase refers “and traditional use”? then fractionated by different solvents … and traditional use? This phrase is unclear. Rapharase or remove “and traditional use”.
Dear reviewer, thank you very much for the comment.” Preparation of Ethnobotanical Use of Hypericum scabrum” was added at the material and methods part. The changes are in a red colour.
- Line 24 – “a slight modification” remove it. It is the detail for the methodology section. Introduction
Dear reviewer, thank you for the comment. “a slight modification” was removed.
- Line 25 – “Essential oils were obtained from the secretory canals of a plant. The canals of the aerial parts, flower, and fruit are defined by the presence of monoterpene hydrocarbons, whereas secretory canals of roots include alkanes.” Essential oils were obtained from plant material (different parts)… not strictly from the secretory canals.
Rather not „The canals” but essential oils of the aerial parts, flower, and fruit are characterized by the presence of monoterpene hydrocarbons, whereas roots include alkanes. Correct or rephrase.
Dear reviewer, thank you for the comment. It was rephrased.
- Line 55 – “perennial” the same information as in line 47. Remove it.
Dear reviewer, thank you for the comment. It was removed.
- Line 54 – “This plant is a….. „ move this phrase at the beginning of this paragraph (line 47)” as presents general information.
Dear reviewer, thank you for the comment. It was moved as the mentioned.
- Line 93 – “The antioxidant activity of H. scabrum herb was higher compared to the Trolox and vitamin C standard and source of it was α-pinene [31].” The high antioxidant activity is mainly related to polyphenols. α-pinene is a weak antioxidant – it has no hydroxyl groups.
Dear reviewer, thank you for the comment. This sentence was revised.
- Results
In my opinion, this presented content in this section should be re-ordered as follows:
Fig. 1-8 and related results description should be at the end of this section (after Table 6) It allows to better transition between presented issues and results. Firstly, phytochemical properties, then anatomy.
Table 1 and 4. Unify the used terms – Methanol or MeOH. Use the same terms through all tables.
Table 1. Two decimal places are enough. Remove (g) after solvent. Add (g) after „Amount”. Remove vertical lines in the table.
Line 109 - The highest yield of essential oil was obtained for fruit, whereas the least for root?
Table 2 – “Powdered amounts” is the amount used for analysis? If yes, remove it. Yield (v/w) is expressed in % (v/w). or mL/g (or 100 g, or kg) of plant material? It is the yield per all powdered amount specified in the table as “Powdered amounts”, recalculate results and express as above - in % (v/w). or mL/g (or 100 g, or kg) of plant material.
Table 4-6 – specify “traditional use” meaning in methodology. This term is misleading. Traditional preparation? Describe how was obtained.
Dear reviewer, thank you for the comment. The revisions were done.
- Discussion
Use the same order of presented issues as in results (following the described above re-order)
Firstly, discussion of phytochemical properties, then anatomy.
Dear reviewers, thank you for the comment. The revisions were done.
- Line 473 – 477 – Why such a procedure was applied? Please, justify.
Why different parts of Hypericum scabrum were not extracted separately in methanol, water n-hexane, dichloromethane, ethyl acetate, and n-butanol? Extraction in methanol allows to separate only compounds soluble in methanol… (more polar), probably less polar compounds were not extracted, so further fractionation is limited only to more polar compounds (soluble in methanol). Extraction, for example with n-butanol, allow separating more polar compounds from the material than those separated by methanol. So, in my opinion extraction should be performed for each solvent separately.
„for 8 hours for 3 days” 8 hours or 3 days? Specify.
Dear reviewer, thank you for the comment. The revisions were done.
- Line 515 – “dry weight” plant material or extract?. Specify.
Dear reviewer, thank you for the comment. The revisions were done.
- Line 545 – any statistical analysis was performed? Experiments were performed in replicates (Table 1-3)?
Dear reviewer, thank you for the comment. The statistical analysis was done for tables 4 and 5 and the Statistical analysis method was added.
-Conclusions
Line 549 – „Hypericum scabrum has a very high antioxidant potential due high presence of a-pinene, undecane and a-thujene it can be used for the pharmaceutical industry.” As described above this statement is very questionable. Even results show that antioxidant activity of essential oil (where a-pinene is the main component of essential oils from aerial parts ) is significantly lower than in extracts. Extracts contains also polyphenols which are responsible for this activity. Consider the analysis of hyperoside and hypericin contents.
Dear reviewer, thank you for the comment. It was revised. We agree about this sentence. We rewrote it as next “Hypericum scabrum has a high presence of a-pinene, undecane and a-thujene it can be used for the pharmaceutical industry”. We agree that antioxidant activity can be connected with presence of other antioxidants of phenolic nature and naphtodianthrones (hypericin etc.) as well. In our next studeis we will plan to estimate naphtodianthrones (hypericin etc.) content etc.
-Line 555 – “EtOAc extract” strictly speaking EtOAc fraction of methanolic extract.
Dear reviewer, thank you for the comment. It was revised.
Reviewer 3 Report
The manuscript "Antimicrobial and antioxidant activities of extracts, traditional use, essential oils, and anatomical properties of Hypericum scabrum L. (Hypericaceae)" is devoted to studying the chemical composition, antiradical and antimicrobial activities of various extracts of Hypericum scabrum L. Methanol, water, ethyl acetate, n-hexane, butanol, dichloromethane were used as the solvents. GC and GC/MS analyses were performed to determine the compositions of obtained extracts. It is a little bit strange, that the authors used the re-extraction method for obtaining extracts in non-polar solvents.
This work is interesting for fundamental science from the metabolomics point of view, and for the development of the using of promising resources of natural bioactive compounds.
I think the manuscript may be published in the Molecules journal after minor revision, which is described below
- Could you explain more clearly in the manuscript, why do you use re-extraction for obtaining sub-extracts, but didn’t use extraction in non-polar solvents?
- Table 5: In the table, the caption states that DPPH radical scavenging activity is expressed in ug/mL. In the caption of the rows, DHHP is expressed in %. It is not clear, what this data means? In addition, it should be noted that inhibition above 70-80% may be out of the linear range of dependence I(%) vs C(ug/mL), and it is can’t be correctly compared with standard solutions.
- The method of determination of polyphenols should be described: Did you evaporate sub-extracts and then dissolve them in methanol or another solvent? Which volumes of aliquots did you use?
- Lines 17-18: “sub extracts” or “sub-extracts”?
- Line 22: If the method for antioxidant activity is specified, it would be better to mention the method for polyphenols determination.
- Lines 27-30: Which organ corresponds to which composition? Maybe, the word “respectively” is needed at the end of this sentence?
- Line 74: “…of Hypericum scabrumthe and…” check the sentence, please.
Author Response
Dear Reviewer,
Thank you for the comments and suggestions which were very helpful to improve MS.
Please, see bellow detailed responces.
- Could you explain more clearly in the manuscript, why do you use re-extraction for obtaining sub-extracts, but didn’t use extraction in non-polar solvents?
Dear reviewer, thank you for the comment. Liquid liquid extraction is one of the methods we use the most. With this method, when we will isolate in the following stages, it facilitates the separation process by bringing together compounds with the same properties. We have used this method in all the studies we have done so far and it has achieved very successful results.
-Table 5: In the table, the caption states that DPPH radical scavenging activity is expressed in ug/mL. In the caption of the rows, DHHP is expressed in %. It is not clear, what this data means? In addition, it should be noted that inhibition above 70-80% may be out of the linear range of dependence I(%) vs C(ug/mL), and it is can’t be correctly compared with standard solutions.
Dear reviewer, thank you for the comment. It was revised. The DPPH radical scavenging activity is expressed in %. All correction were done.
-The method of determination of polyphenols should be described: Did you evaporate sub-extracts and then dissolve them in methanol or another solvent? Which volumes of aliquots did you use?
Dear reviewer, thank you for the comment. It was revised. Yes. It was done evaporation of sub-extracts and then they were dissolved in methanol. The sub-extracts, extracts were evaporated and then dissolved with 2 ml of methanol.
25-Lines 17-18: “sub extracts” or “sub-extracts”?
Thank you for the comment. The correction was done. It is “sub-extracts”.
26-Line 22: If the method for antioxidant activity is specified, it would be better to mention the method for polyphenols determination.
Thank you for the suggestion. The method of polyphenol determination was specified. Please, see it in the text in a red colour.
Total phenolic estimation
Total phenolics were determined by using Folin–Ciocalteu reagent. The sub-extracts, extracts were evaporated and then dissolved with 500 μL of 70% methanol (HPLC-Gradient grade, VWR chemicals) at 70 °C. The mixtures were centrifuged at 3500g for 10 min and the supernatants were collected in separate tubes. The pellets were re-extracted under identical conditions. Supernatants were combined and used for total phenolics assay. For total phenolics assay 20 μL of extract was dissolved into 2 mL of distilled water. 200 μL of dissolved extract was mixed with 1 mL of Folin–Ciocalteu reagent (previously diluted tenfold with distilled water) and kept at 25 °C for 3–8 min; 0.8 mL of sodium bicarbonate (75 g L−1) solution was added to the mixture. After 60 min at 25 °C, absorbance was measured at 765 nm. The results were expressed as gallic acid equivalents.
Complete extraction of phenolic compounds is the next critical step after sample preparation. The most common techniques to extract phenolics employ solvents, either organic or inorganic. Several parameters may influence the yield of phenolics, including extraction time, temperature, solvent-to-sample ratio, the number of repeat extractions of the sample, as well as solvent type. Furthermore, the optimum recovery of phenolics is different from one sample to the other and relies on the type of plant and its active compounds. The highest levels of phenolics are extracted from Vitis vinifera wastes and sunflower meal using pure methanol (Casazza et al., 2010). These differences could be due to the properties of the phenolic components of the plants concerned.
Casazza A.A., Aliakbarian B., Mantegna S., Cravotto G., Perego P. Extraction of phenolics from Vitis vinifera wastes using non-conventional techniques. J. Food Eng. 2010;100:50–55. doi: 10.1016/j.jfoodeng.2010.03.026.
- Lines 27-30: Which organ corresponds to which composition? Maybe, the word “respectively” is needed at the end of this sentence?
Dear Editor and reviewers, thank you for the comment. It was revised.
- Line 74: “…of Hypericum scabrumthe and…” check the sentence, please.
Thank you for the suggestion. The sentence was corrected. It is “Just in 2020, the scientists found three new compounds with untypical skeletons were segregated from the air-dried aerial parts of Hypericum scabrum plant. “ The changes in a red colour.
Round 2
Reviewer 1 Report
Some necessary corrections were made and the manuscript was improved. In my opinion, the current version meets the requirements for publication.
Author Response
Dear Reviewer 1,
Thank you for the comments and suggestions. We tried our best to improve MS and your comments were helpful.
Sincerely yours
Reviewer 2 Report
Dear Authors, Thank you for the changes within the manuscript and their improvement.
However, I must point out the incorrect expression of sentences - see my previous comment:
Many sentences do not make sense and do not even have a sentence - see eg L69-71, L. 95-96 (the verb is completely missing ...). Again, I give only examples, but in many places, the text is incomprehensible or perhaps misleading. It must be revised, rewritten.
This comment was not taken into account when editing the text. Please note that the two sentences I point out are just an example. It is necessary to read the manuscript in its entirety and avoid similar problematic sentences.
Author Response
Dear Reviewer 2,
Thank you for the comment. Please, see response for the comment bellow
- However, I must point out the incorrect expression of sentences - see my previous comment:
Many sentences do not make sense and do not even have a sentence - see eg L69-71, L. 95-96 (the verb is completely missing ...). Again, I give only examples, but in many places, the text is incomprehensible or perhaps misleading. It must be revised, rewritten.
This comment was not taken into account when editing the text. Please note that the two sentences I point out are just an example. It is necessary to read the manuscript in its entirety and avoid similar problematic sentences.
The text and MS was revised. A sentence L69-71, L. 95-96 were revised. We corrected MS and changes in a purple colour. The text was checked according to the https://www.grammarly.com/.